# Benzimidazole-2-Phenyl-Carboxamides as Dual-Target Inhibitors of BVDV Entry and Replication

**DOI:** 10.3390/v14061300

**Published:** 2022-06-14

**Authors:** Roberta Ibba, Federico Riu, Ilenia Delogu, Ilenia Lupinu, Gavino Carboni, Roberta Loddo, Sandra Piras, Antonio Carta

**Affiliations:** 1Department of Medical, Surgical and Experimental Sciences, University of Sassari, 07100 Sassari, Italy; ribba@uniss.it (R.I.); friu1@uniss.it (F.R.); ilenialup@gmail.com (I.L.); acarta@uniss.it (A.C.); 2Department of Biomedical Sciences, Cittadella Universitaria Monserrato, University of Cagliari, 09042 Monserrato, Italy; deloguilenia@gmail.com; 3Department of Biomedical Sciences, University of Sassari, 07100 Sassari, Italy; g.carboni43@studenti.uniss.it

**Keywords:** benzimidazole-2-phenyl-carboxamides, BVDV, dual target, antivirals, entry inhibition, replication inhibition

## Abstract

Bovine viral diarrhea virus (BVDV), also known as Pestivirus A, causes severe infection mostly in cattle, but also in pigs, sheep and goats, causing huge economical losses on agricultural farms every year. The infections are actually controlled by isolation of persistently infected animals and vaccination, but no antivirals are currently available to control the spread of BVDV on farms. BVDV binds the host cell using envelope protein E2, which has only recently been targeted in the research of a potent and efficient antiviral. In contrast, RdRp has been successfully inhibited by several classes of compounds in the last few decades. As a part of an enduring antiviral research agenda, we designed a new series of derivatives that emerged from an isosteric substitution of the main scaffold in previously reported anti-BVDV compounds. Here, the new compounds were characterized and tested, where several turned out to be potent and selectively active against BVDV. The mechanism of action was thoroughly studied using a time-of-drug-addition assay and the results were validated using docking simulations.

## 1. Introduction

Bovine viral diarrhea virus (BVDV) is a single-stranded positive-sense RNA virus that belongs to the Pestivirus genus of the Flaviviridae family, recently renamed Pestivirus A in the 2017 report from the International Committee on Taxonomy of Viruses [1]. The Pestivirus genus also contains more animal pathogens, such as the border disease virus (BDV) and the classical swine fever virus (CSFV) [2]. Not only can cows be the host of BVDV, but pigs, sheep, goats and other wild and domestic ruminants can also be infected [3]. BVDV causes teratogenesis, abortion, early embryonic death, immune system and respiratory disorders in cows, resulting in acute infections of immunocompetent cattle, giving rise to a mortality rate ranging from approximately 20 to 30% [4] and an estimated economical loss of approximately 10 and 40 million dollars per million calvings. Newborns can be infected through the placenta, leading to the presence of cattle persistently infected (PI) with BVDV on farms [5]. BVDV was also detected as a troublesome pollutant in commercially available lots of fetal bovine serum and cell lines commonly used in the laboratories [6], and therefore, was revealed as a contaminant in interferons and vaccines for medical use [7,8].

No antivirals are currently available for controlling BVDV infections in laboratories or on farms; the vaccination protocols and the isolation of PI animals is the only validated strategy currently in use on farms to limit the transmission and reduce the economic loss [9]. BVDV continues to cause agronomical trouble; therefore, the identification of molecules that are capable of binding and inhibiting the virus replication and transmission remains a major antiviral research goal.

BVDV single-stranded RNA is translated by the host cell into a polyprotein, which is processed by viral and host proteases and cleaved into four structural (Protein C, Erns, E1 and E2) and eight functional proteins [10]. E2 glycoprotein binds the cell-surface receptor (CD46 or CD81) and causes the membrane fusion to begin, which allows the BVDV single-stranded RNA to be released in the host cytoplasm [2,11]. The non-structural viral proteins have been the most inhibited in drug discovery in the last few decades, such as NS5B RNA-dependent RNA-polymerase (RdRp), NS4a protease and NS3 helicase, while only recently, the structural proteins were targeted in order to inhibit the early stages of virus infection [12].

In recent years, many selective anti-BVDV compounds were reported, with a virus-targeting or host-targeting approach, inter alia viral polymerase ligands [13,14,15,16], protease inhibitors [17] and human cellular enzymes targeting compounds [18,19,20,21]. The antiviral research against Pestivirus genus pathogens is still a challenge. Among others, benzimidazole derivatives were demonstrated to be active against RNA viruses in the last few decades [22,23,24,25,26,27].

As a part of our enduring antiviral research agenda, a series of angular and linear *N*-polycyclic derivatives active against BVDV, HCV and other related viruses were designed and synthetized [24,28,29]. The viral target for BVDV inhibition was ascertained in the RNA-dependent RNA-polymerase (RdRp) [30,31]. Aiming to select the best scaffold endowed with antiviral activity, we synthesized several polycyclic heteroaromatic derivatives, in turn, improving selectivity and potency [29,32,33] The most promising anti-BVDV compounds that emerged possess the halo-benzotriazole-2-phenyl carboxamide scaffold. Based on the bioisostere theory and the literature on benzimidazole scaffold, we designed the isosteric substitution of the N2 of benzotriazole derivatives depicted in Figure 1 with a carbon atom, obtaining benzimidazole derivatives. This variance limited the free rotation of the phenyl moiety bound to the atom in position 2 and transformed the nitrogen in position 1 from an H-bond acceptor to an H-bond donor. This resulted in a series of benzimidazole derivatives (**1**–**7n**, **1**–**7o**, **1**–**7p**) wherein substitution based on benzimidazole moiety was evaluated; meanwhile, the substitution of R’ = 4-Cl, 4-NO_2_ or 3,4,5-trimethoxy was conserved.

## 2. Materials and Methods

### 2.1. Chemistry

#### 2.1.1. General Synthetic Strategies

All starting materials were purchased from Sigma-Aldrich (Merck KGaA, Darmstadt, Germany), Across Organics (Geel, Belgium) and Carlo Erba (Milan, Italy) producers. Nitrogen group reduction to amine derivatives was performed in ethanol using three different routes as follows: chlorine derivatives reductions were made with methylhydrazine as a bland reducing agent in ethanol using the autoclave at 100 °C. All the other reductions were conducted in ethanol with Pd/C as a catalyst and H_2_ at room temperature, or with Pd/C and hydrazine at 80 °C. Benzimidazole ring closure was carried out as described by Bahrami et al. [34] by mixing o-phenylenediamines and aldehydes in a ratio of 1:1 in acetonitrile as the solvent, with H_2_O_2_ 30% (ratio 1:7) and HCl 37% (ratio 1:3.5) added at room temperature for the proper number of hours. Amide groups were obtained from the reaction of aniline derivatives with benzoyl chloride derivatives in a ratio of 1:1.2 in DMF at 80 °C till the reaction was completed. Products were purified via crystallization from ethanol or methanol, or via flash chromatography using proper elution by mixing suitable chosen solvents among petroleum ether (PE), diethyl ether (DE), ethyl acetate (EA), chloroform and methanol.

#### 2.1.2. Chemical Characterization

Nuclear magnetic resonance (NMR) spectra were recorded with a Bruker Avance III 400 NanoBay (400 MHz) instrument from Bruker (Billerica, MA, US) from sample solutions in deuterated solvents, Acetone or DMSO. ^1^H NMR chemical shifts are reported in parts per million (ppm) downfield from tetramethylsilane (TMS) used as an internal standard. Chemical shift values are reported in ppm (*δ*) and coupling constants (*J*) are reported in hertz (Hz). Signal multiplicities are represented as s (singlet), ws (wide singlet), d (doublet), dd (doublet of doublets), ddd (doublet of doublet of doublets), t (triplet), td (triplet of doublets), q (quadruplet) and m (multiplet). The assignment of exchangeable protons (OH and NH) was confirmed via the addition of D_2_O. ^13^C NMR chemical shifts are reported downfield from tetramethylsilane (TMS) used as an internal standard. A suitable method among the APT (attached proton test) and jmod (J-modulated spin-echo for X-nuclei coupled to H-1 to determine the number of attached protons) was selected for each compound. Two-dimensional NMR experiments HSQC (heteronuclear single quantum coherence) and HMBC (heteronuclear multiple bond correlation) were performed to correctly assign the peaks. The solutions for ESI-MS measurements were prepared at a concentration of 1.0–2.0 ppm by dissolving and serially diluting the solid compounds in HPLC acetonitrile. Mass spectra in the positive-ion mode were obtained on a Q Exactive Plus Hybrid Quadrupole-Orbitrap (Thermo Fisher Scientific, Waltham, MA, USA) mass spectrometer. The solutions were infused at a flow rate of 5.00 μL/min into the ESI chamber. The spectra were recorded in the *m/z* range of 150–800 at a resolution of 140,000 and accumulated for at least 2 min to increase the signal-to-noise ratio. The instrumental conditions used for the measurements were as follows: spray voltage 2300 V, capillary temperature 250 °C, sheath gas 10 (arbitrary units), auxiliary gas 3 (arbitrary units), sweep gas 0 (arbitrary units) and probe heater temperature 50 °C. ESI-MS spectra were analyzed by using Thermo Xcalibur 3.0.63 software (Thermo Fisher Scientific, Waltham, MA, USA), and the average deconvoluted monoisotopic masses were obtained through the Xtract tool integrated with the software. Each compound melting point (m.p.) was taken in open capillaries in a Digital Electrothermal melting point apparatus and was uncorrected. Retention factors (R_f_) were measured via thin-layer chromatography (TLC) using Merck F-254 commercial plates (Merck KGaA, Darmstadt, Germany).

### 2.2. Biology

#### 2.2.1. Cell Lines and Viruses

Cell lines were purchased from the American Type Culture Collection (ATCC). The absence of mycoplasma contamination was periodically checked using the Hoechst staining method. The cell line supporting the multiplication of BVDV was the following: Madin-Darby Bovine Kidney (MDBK) (ATCC CCL 22 (NBL-1) Bos Taurus). The virus was purchased from American Type Culture Collection (ATCC), Bovine Viral Diarrhoea Virus (BVDV) (strain NADL (ATCC VR-534)).

#### 2.2.2. Cytotoxicity Assay

Cytotoxicity assays were run in parallel with antiviral assays. MDBK cells were seeded at an initial density of 6 × 10^5^ cells/cm^3^ in 96-well plates in culture medium (Minimum Essential Medium with Earle′s salts (MEM-E) with L-glutamine, supplemented with 10% horse serum and 1 mM sodium pyruvate, and 0.025 g/L kanamycin). Cell cultures were then incubated at 37 °C in a humidified, 5% CO_2_ atmosphere in the absence or presence of serial dilutions of test compounds. The cell viability was determined after 48–96 h at 37 °C using the 3-(4,5-dimethylthiazol-2-yl)-2,5-diphenyltetrazolium bromide (MTT) method [35]. The MTT data were processed using the statistical method of linear regression.

#### 2.2.3. Antiviral Assay

Compound activity against BVDV was based on the inhibition of virus-induced cytopathogenicity in MDBK cells acutely infected with an m.o.i. of 0.01. Briefly, MDBK cells were seeded in 96-well plates at a density of 3 × 10^4^ cells/well and were allowed to form confluent monolayers by incubating overnight in a growth medium at 37 °C in a humidified CO_2_ (5%) atmosphere. Cell monolayers were then infected with 50 mL of virus dilution in a maintenance medium (MEM-E with L-glutamine, supplemented with 0.5% inactivated FBS, 1 mM sodium pyruvate and 0.025 g/L kanamycin) to give an m.o.i. of 0.01. After 2 h, 50 mLof maintenance medium, without or with serial dilutions of test compounds, was added. After 3 days of incubation at 37 °C, cell viability was determined using the MTT method. Linear regression analysis: viral and cell growth at each drug concentration is expressed as a percentage of untreated controls and concentrations resulting in 50% (EC_50_ and CC_50_) growth inhibition. NM 108 (2′-*C*-methylguanosine) and ribavirin were used as positive controls. The selectivity index (S.I.) was used, which is a parameter of preferential antiviral activity of a compound related to its cytotoxicity (CC_50_/EC_50_). Each experiment was conducted in duplicate (two wells in parallel) and the experiments were performed in three copies.

#### 2.2.4. Time of Drug Addition Assay

A time-of-addition experiment was carried out with MDBK cells. The confluent monolayers of MDBK cells, seeded in 24-well tissue culture plates, were infected for 1 h at room temperature with 6650 PFU of BVDV, corresponding to a multiplicity of infection of 3 PFU/cell. After adsorption for 60 min, the monolayers were washed two times with a maintenance medium in the presence of HS inactivated and incubated with the same medium at 5% CO_2_ and 37 °C. The test medium containing 10 × EC_50_ compound concentration was added at –1 to 0 (pretreatment), 0 to 2 (during infection), 2 to 4, 4 to 6, 6 to 8, 8 to 10, 10 to 12, 12 to 14 and 14 to 16. After each incubation period, the monolayers were washed two times with a maintenance medium and incubated with a fresh medium until 12 h post-infection. Then, the monolayers were frozen at −80 °C and the viral titers were determined using a plaque reduction assay.

#### 2.2.5. Virucidal activity

A suspension of viral particles was directly exposed to each test compound and incubated for 1 h at 4 °C. Then, the mixtures were diluted in series and used to infect the cells; the infectious titers were recorded and compared with those obtained with an untreated viral suspension.

### 2.3. Docking

AutoDock Vina [36] was selected as the docking program. As a starting point, the crystal structure of BVDV1 envelope glycoprotein E2 was retrieved from the RCSB Protein Data Bank (rcsb.org, PDB code 2YQ2) [22]. NAG (2-acetamido-2-deoxy-β-d-glucopyranose) and water molecules were removed from the co-crystallized structure.

Compounds **2c**, **2d**, **2e**, **6c** and **7c**, were first minimized through Avogadro [37] version 1.1.1 and then saved as PDB files. Protein and ligand structures were processed through AutoDockTools (ADT) [38] version 1.5.6, generating the PDBQT files and obtaining the coordinate and volume information for the docking grid box: 30–30–30 Å; x = –17.095, y = 35.857 and z = –38.325. For the ligands′ structure, all rotatable bonds were made flexible, including the amide bonds. Docking was carried out by making some amino acids flexible, viz., Ser57, Thr60, Arg61, Gln87, Glu89, Asp91, Pro105, Val153 and Arg154. These amino acids were exposed in the target binding pocket of the protein and shown to stabilize the structural conformation of different literature compounds that docked in that precise site [12]. As docking parameters, the exhaustiveness value was set to 64 and the number of poses to 10 for each docking prediction. The affinity energy range was set to be ≤ 2 kcal·mol^−1^ above that of the best binding pose for each ligand. For the evaluation of the affinity prediction, the best-docked pose of each investigated ligand was taken and visualized in PyMOL 2.3.4 [39]. The docking studies were conducted on a PC with an Intel^®^ Core™ i7–9705H CPU @ 2.60 GHz with 8 GB RAM running Ubuntu 16.04 as an operating system. The RdRp crystal structure was retrieved by RSCB PDB (PDB code 1S48). The grid box was set to 18–18–22 Å (spacing 1 Å) around the coordinating point with x/y/z values of –28.322/157.375/22.518.

## 3. Results and Discussion

### 3.1. Chemistry

Derivatives **1**–**7c**, **1**–**7d** and **1**–**7e** were obtained as described in Figure 1. Intermediate *o*-phenylenediamines **1**–**7** (commercially purchased) were mixed with 4-nitrobenzaldehyde **8a** in a ratio of 1:1 to obtain nitro-derivatives **1**–**7a** that were appropriately reduced to aniline derivatives **1**–**7b**. Derivatives **1**–**7c**, **1**–**7d** and **1**–**7e** were gained via the reaction of aniline derivatives **1**–**7b** with the corresponding benzoyl chloride derivative (**9c**,**d**,**e**) in a ratio of 1:1.2 in DMF at 80 °C till reaction was completed.

### 3.2. Anti-BVDV Activity

Concerning the benzimidazole derivatives (**1**–**7c**, **1**–**7d**, **1**–**7e**), which were synthesized and evaluated for activity and selectivity against BVDV, the results are reported in Table 1. It clearly shows that when R’ is a chlorine atom (**1**–**7e**), the corresponding compounds had no relevant antiviral activity and the EC_50_ values were always higher than CC_50_ ones. All trimethoxy-derivatives (**1**–**7c**) were found to be highly potent BVDV inhibitors with EC_50_ values ranging between 0.09 and 41 µM, whereas only some nitro-derivatives (**2**–**5d**) showed interesting EC_50_ values: 1.9, 53, 7.9 and 3.6 µM, respectively. Selecting the most active trimethoxy-derivatives (**1**–**7c**), preliminary SARs highlighted that the substitution on the benzimidazole scaffold in position 5 or both 5 and 6 generally increased the activity, and the presence of one (**2c**) or two chlorine atoms (**3c**) strengthened the activity but was coupled with cytotoxicity, with CC_50_ values of 45 and 28 µM, respectively. The presence of one or two smaller atoms, such as fluorine (**4c** and **5c**), augmented the activity (EC_50_: 41 and 1.4 µM) with cytotoxicity values always higher than 100 µM. A valuable anti-BVDV activity was obtained when a methyl group was in position 5 or both 5 and 6 on the benzimidazole main scaffold, resulting in derivatives **6c** and **7c**, with EC_50_ values of 0.23 and 0.3 µM, respectively. This SAR analysis was confirmed by the selectivity index (S.I.) values, also reported in Table 1. The three most potent compounds, namely, **2c**, **6c** and **7c**, also showed the highest S.I. values. Several new compounds showed a better score than the positive control compounds, namely, NM108 (S.I. 66.7) and the gold-standard reference ribavirin, for which the S.I. was 5.6.

### 3.3. Time of Drug Addition

According to the discussed results, the three most promising compounds, namely, **2c**, **6c** and **7c**, were selected to deepen the study of the mechanism of virus inhibition for this new class of compounds. A time of drug addition assay was performed to identify the time point in the virus replication cycle where the new compounds exerted the highest antiviral activity, and the results are depicted in Figure 2.

A time of drug addition assay indicates the time at which the compound reaches the maximum antiviral activity; thus, knowing the virus life cycle, the virus replication step inhibited by the analyzed compound can be identified. The results showed that the pretreatment with the three tested compounds resulted in a slight reduction in virus replication; the same poor activity was detected between 10- and 16-h post-infection. The best score was detected when cells were treated at the infection moment. The poor activity detected in pretreated cells and the highest potency when the cells were treated simultaneously with compounds and BVDV suggested that benzimidazole compounds inhibited the early stage of the virus infection of the host cell. Moreover, these data let us suppose that the target lay on the virus envelope rather than on the cells since no cell protection during pretreatment was detected.

Chlorinated compound **2c** showed a consistent antiviral activity along the viral replication cycle more distinctly than derivatives **6c** and **7c**. This suggested that a non-structural protein was also targeted by the present series of compounds; indeed, they were designed and synthesized to target and inhibit the viral RdRp.

The time of drug addition output showed that the three tested compounds turned out to be the most active in the early stage of infection. The host-targeting effect was ruled out by the poor results of the cell pretreatment. The viral-targeting effect may result in virucidal or virustatic activity. The former was evaluated.

### 3.4. Virucidal Activity

To detect the antiviral mechanism of action, the virucidal activity of the three selected compounds, namely, **2c**, **6c** and **7c**, was investigated. BVDV was incubated at 4 °C for 1 h; the mixture was then used to infect cells. It resulted in infected cultures of host cells that were almost comparable to the untreated but infected ones, as shown in Figure 3. The single treatment of virus or cell pretreatment did not result in the highest antiviral activity, while it was demonstrated that when the three components, namely, virus, cell and ligand, were mixed at the same time, the compounds inhibited the receptor binding.

### 3.5. Molecular Docking

#### 3.5.1. Trimethoxy-Substituted Compounds **2c**, **6c** and **7c** Docked in a Strategic Binding Pocket in E2 Protein

BVDV E2 glycoprotein was demonstrated to have a crucial role in cell entry [40]. Docking predictions were carried out to evaluate the correlation between the biological results and the predicted mode of binding of the different ligands. Considering the biological assessment, compounds **2c**, **6c** and **7c** were reported as the hit compounds endowed with the best antiviral activity of the present series of compounds. The docking predictions provided a value of affinity energy for each ligand targeting the same protein binding pocket. The tri-methoxy derivatives **2c**, **6c** and **7c** showed comparable affinity energies between −6.4 and −6.9 kcal·mol^−1^, as reported in Table 2.

More in-depth analyses of the protein–ligand interactions are reported in the following discussion, involving the polar and non-polar contacts between the ligands and a specific portion of the BVDV glycoprotein E2 (PDB ID: 2YQ2). Starting with compound **2c** reported in Figure 4, its best-predicted pose showed that the molecule establishing different polar contacts with some exposed amino acids of the binding pocket of E2 glycoprotein. The docking of the ligands to the glycoprotein E2 was performed on a specific region located in the DB domain (residues 88–164) of E2 glycoprotein, one of the most distal portions from the virus membrane, and hence, one of the most exposed on the viral surface. This specific portion is often targeted by anti-BVDV compounds [40]. The methoxy groups have a crucial role: the oxygen of OCH_3_ in the C–3 position accepts a hydrogen bond with the intrinsic NH of Leu103. The side-chain isopropyl CH_3_ of Leu103 has a polarized C–H bond, which can interact with the benzimidazole nitrogen atom of compound **2c** (3.2 Å length). The phenolic hydroxyl group of Tyr156 contacts the C–4 methoxy group of the ligand. Moreover, the best-predicted conformation of the benzimidazole scaffold played a crucial role, with two polar contacts (2.3 and 2.9 Å length) within the acceptor carbonyl oxygen of Arg58. This showed how the polar term was fundamental for E2–**2c** affinity, suggesting a strong binding of compound **2c** and explaining its good experimental antiviral profile. Regarding the hydrophobic interactions involving compound **2c**, the tri-methoxy phenyl moiety was the most-involved portion: it was surrounded by different amino acids: Gly102, Tyr156, Val93, Phe99, Arg154 and Phe101. The middle phenyl-amide moiety interacted with Gly102, but also Ser57, Cys104 and Pro105. Furthermore, the 5-chlorobenzimidazole scaffold had nonpolar contacts with Pro105, Cys106, Arg58, Asp107, Leu103 and Cys104.

Compound **6c** showed a characteristic affinity with the target, where the polar protein–ligand interactions involved each moiety of the ligand. The OCH_3_ in the C–3 position established an H-bond with the guanidine NH_2_ group of Arg154. As previously discussed for **2c**, compound **6c** interacted with Leu103, involving a polar interaction between the benzimidazole NH of the ligand and the carbonyl oxygen of the leucine residue (2.6 Å bond length). In addition, the carbonyl oxygen of the amide linker accepted a hydrogen bond 2.6 Å in length from the side-chain hydroxyl group of Ser57. The great antiviral activity of compound **6c** was shown through the biological assessment and it was confirmed here by this docking prediction (Figure 5A), where the electrostatic term seemed crucial for the binding. Different hydrophobic interactions were established within neighboring amino acids: of particular interest were Tyr156, Arg154 and Leu103 on the benzimidazole scaffold side; Arg58, Ser57 and Pro105 for the phenyl-amide middle portion; and Leu103, Pro105, Asp91 and Cys106 for the tri-methoxy-phenyl moiety.

To complete the series of tri-methoxy-derived compounds, derivative **7c** resulted in a good antiviral agent but was slightly less potent than compounds **2c** and **6c**. The docking results showed that derivative **7c** had a unique H-bond with a methionine amid the benzimidazole NH of the ligand and the carbonyl oxygen of Met141 (2.5 Å length) of the E2 target. The same amino acid also established nonpolar interactions with the benzimidazole scaffold, as was the case for Ala143 and Lys109. As can be seen in Figure 5B, the hydrophobic term was crucial for the protein–ligand relationship. Furthermore, Leu142 and Asp107 set hydrophobic interactions with the phenyl-amide portion, while Arg61, Cys106 and Asn144 contacted the tri-methoxy-phenyl moiety.

#### 3.5.2. Trimethoxy-Substituted Compounds **2c**, **6c** and **7c** Docked in a Strategic Binding Pocket in E2 Protein

The 5-chloro derivatives **2c** (tri-methoxy), **2d** (*p*-NO_2_) and **2e** (*p*-Cl) were used for deeper docking simulations in order to compare the prediction outputs with the experimental biological results and to better understand the different interactions of the R_3_, R_4_ and R_5_ substituents within the E2 glycoprotein target. The binding profile of compound **2c** was previously described. The high number of polar–nonpolar contacts gave a tight binding of the molecule to the binding pocket surface. As for the *p*-nitro derivative **2d**, it interacted with E2 protein through some nonpolar interactions, while it did not show hydrogen bonds with neighboring polar amino acids (Figure 6A). Regarding compound **2e**, depicted in its best-binding pose in Figure 6B, it established a unique H-bond: the amidic oxygen of the ligand interacts as an acceptor with the side-chain NH of Arg154. Few hydrophobic interactions contributed to its interaction with E2 protein. In general, we could assume that the weaker antiviral effect of compounds **2d** and **2e** in comparison to **2c** could be predictively confirmed by these docking studies. In general, analyzing the polar interactions between this class of molecules and the E2 binding pocket, Leu103 and Arg154 amino acids were predicted to be crucial for the ligand binding and were important for their conformational stabilization of their protein-bound state.

#### 3.5.3. Docking of Compound **2c** to BVDV RNA-Dependent RNA Polymerase (RdRp)

In order to collect in silico data regarding its mechanism of action, the binding to the envelope glycoprotein E2 or the RNA-dependent RNA polymerase (RdRp) of the benzimidazole-based lead compound **2c** was compared. The favorable binding of compound **2c** to the binding site in the envelope protein E2 is reported below. The binding to the RdRp of BVDV was evaluated, considering the most important amino acids for the docking to this protein, which was previously reported by us and other researchers in the literature [29,30,31].

The affinity score of the binding between the best-docked pose of derivative **2c** and the RdRp binding site was −5.9 kcal·mol^−1^, with average energy between the first 10 most-ranked poses of ~ −5.3 kcal·mol^−1^. The binding of compound **2c** was better predicted to occur at the E2 active site in comparison to that at RdRp superficial site, considering the affinity scores.

Moreover, regarding the protein–ligand interactions, the importance of the polar and nonpolar contacts between compound **2c** and the envelope E2 protein were previously-reported. In contrast, the contact between ligand **2c** and the superficial site of RdRp was predicted not to create polar contacts, not showing any H-bond with the exposed amino acids. The ligands had different hydrophobic interactions with the RdRp site. The 5-chlorobenzimidazole scaffold interacted with Asp378, Thr379, Gly251, Asp382 and Lys375. The remaining moieties of the ligand interacted with Glu363, Gln359 and Leu360. The results are depicted in Figure 7.

We concluded that ligand **2c**, in its best-predicted pose, showed an interesting pattern of electrostatic and hydrophobic interactions with the exposed amino acids of the envelope E2 protein. In contrast, when the best-docked pose of ligand **2c** approached the superficial pocket of RdRp, its binding profile appeared weaker in terms of the affinity energy score and polar/nonpolar interactions.

## 4. Conclusions

We undertook a bioisosteric substitution of benzotriazole scaffold of our previously reported anti-BVDV compounds with benzimidazole scaffold. The resulting 21 compounds **1**–**7c**, **1**–**7d** and **1**–**7e** were tested against BVDV and MDBK cell lines and selectivity indexes were also calculated. Three of them were found to be active in the low micromolar range, and **2c** showed the highest EC_50_ value of 90 nM. It was found to be active and potent in the early stages of infection, inhibiting the entry of the virus into the host cell and interfering with the receptor binding, whilst no direct virucidal activity was detected. Compound **2c** showed detectable activity, also between 2 and 14 h post-infection, when viral replication is ongoing. The key protein in the cell surface recognition was the viral E2 glycoprotein and our hit compound 2c was shown to bind the target pocket with an affinity energy value of −6.4 kcal·mol^−1^. We concluded that ligand **2c**, in its best-predicted pose, showed an interesting pattern of electrostatic and hydrophobic interactions with the exposed amino acids of the envelope E2 protein. In contrast, when the best-docked pose of ligand **2c** approached the superficial pocket of RdRp, its binding profile appeared weaker in terms of affinity energy score and polar/nonpolar interactions. These results appeared in agreement with the biological outputs. Further studies should be carried out to validate these results.

## 5. Experimental Characterization

### 5.1. Characterization of Intermediates **1**–**7a** and **1**–**7b**

#### 5.1.1. 2-(4-Nitrophenyl)-1H-benzo[d]imidazole (**1a**)

Compound **1a** (C_13_H_9_N_3_O_2_, MW 239.229) was obtained with a total yield of 44%; m.p. 214–215 °C; TLC (CHCl_3_/CH_3_OH 95/5): R_f_ 0.72. ^1^H-NMR (DMSO-*d_6_*): *δ* 8.49 (4H, s, H-2′,3′,5′,6′), 7.76 (2H, m, H-4,7), 7.41 (2H, m, H-5,6). ^13^C-NMR (jmod, DMSO-*d_6_*): δ 148.45 (C), 148.24 (C), 136.98 (2C), 133.63 (C), 128.10 (2CH), 124.41 (2CH), 124.18 (2CH), 115.18 (2CH). ESI-MS (*m/z*): calcd. for C_13_H_9_N_3_O_2_ 240.077, found 240.077 [M+H]^+^.

#### 5.1.2. 5-Chloro-2-(4-nitrophenyl)-1H-benzo[d]imidazole (**2a**)

Compound **2a** (C_13_H_8_ClN_3_O_2_, MW 273.674) was obtained with a total yield of 97%; m.p. 256–258 °C; TLC (PE/ES 7/3): R_f_ 0.44. ^1^H-NMR (DMSO-*d*_6_): *δ* 8.43 (4H, s, H-2′,3′,5′,6′), 7.73 (1H, d, J = 1.6 Hz, H-4), 7.69 (1H, d, J = 8.4 Hz, H-7), 7.308 (1H, dd, ^1^J = 10.4 Hz, ^2^J = 2 Hz, H-6), 3.68 (1H, s, NH). ^13^C-NMR (DMSO-*d*_6_): *δ* 150.30 (C), 148.07 (C), 140.27 (C), 137.78 (C), 135.25 (C), 127.62 (2CH), 127.36 (C), 124.31 (2CH), 123.39 (CH), 116.70 (CH), 115.27 (CH). ESI-MS (*m/z*): calcd. for C_13_H_8_ClN_3_O_2_ 274.038, found 274.038 [M+H]^+^.

#### 5.1.3. 5,6-Dichloro-2-(4-nitrophenyl)-1H-benzo[d]imidazole (**3a**)

Compound **3a** (C_13_H_7_Cl_2_N_3_O_2_, MW 308.120) was obtained with a total yield of 69%; m.p. 226 °C; TLC (PE/EA 7/3): R_f_ 0.58. ^1^H-NMR (DMSO-*d_6_*): δ 8.40 (4H, ws, H-2′,3′,5′,6′), 7.91 (2H, s, H-4,7). ^13^C-NMR (jmod, DMSO-*d_6_*): δ 151.50 (C), 148.25 (C), 188.80 (2C), 134.88 (C), 127.42 (2CH), 125.47 (2C), 124.32 (2CH), 116.83 (2CH). ESI-MS (*m/z*): calcd. for C_13_H_7_Cl_2_N_3_O_2_ 307.999, found 307.999 [M+H]^+^.

#### 5.1.4. 5-Fluoro-2-(4-nitrophenyl)-1H-benzo[d]imidazole (**4a**)

Compound **4a** (C_13_H_8_FN_3_O_2_, MW 257.220) was obtained with a total yield of 87%; m.p. 226.3 °C; TLC (PE/EA 7/3): R_f_ 0.45. ^1^H-NMR (DMSO-*d_6_*): δ 8.45 (4H, d, J = 1.6 Hz, H-2′,3′,5′,6′), 7.74 (1H, m, H-7′), 7.45 (1H, m, H-4′), 7.23 (1H, m, H-6′). ^13^C-NMR (jmod, DMSO-*d_6_*): δ 160.54 (C), 158.18 (C), 149.72 (C), 148.35 (C), 137.91 (C), 133.97 (C), 127.95 (2CH), 124.36 (2CH), 116.44 (CH), 112.26 (CH), 101.29 (CH). ESI-MS (*m/z*): calcd. for C_13_H_8_FN_3_O_2_ 258.067, found 258.067 [M+H]^+^.

#### 5.1.5. 5,6-Difluoro-2-(4-nitrophenyl)-1H-benzo[d]imidazole (**5a**)

Compound **5a** (C_13_H_7_F_2_N_3_O_2_, MW 275.210) was obtained with a total yield of 52%; m.p. 292 °C; TLC (PE/EA 7/3): R_f_ 0.40. ^1^H-NMR (DMSO-*d*_6_): *δ* 8.39 (4H, m, H-2′,3′,5′,6′), 7.715 (2H, t, H-4,7). ^13^C-NMR (jmod, DMSO-*d*_6_): *δ* 149.70 (C), 148.88 (C), 148.39 (2C, dd, ^1^J_C-F_ = 245 Hz, ^2^J_C-F_ = 19 Hz, 2C-F), 131.84 (C, m), 131.66 (C), 131.25 (C, m), 128.35 (2CH), 124.28 (2CH), 102.90 (2C, dd, ^2^J_C-F_ = 15 Hz, ^3^J_C-F_ = 8 Hz, CH-4,7). ESI-MS (*m/z*): calcd. for C_13_H_7_F_2_N_3_O_2_ 276.058, found 276.058 [M+H]^+^.

#### 5.1.6. 5-Methyl-2-(4-nitrophenyl)-1H-benzo[d]imidazole (**6a**)

Compound **6a** (C_14_H_11_N_3_O_2_, MW 253.256) was obtained with a total yield of 42%; m.p. 94–95 °C; TLC (CHCl_3_/CH_3_OH 95/5): R_f_ 0.74. ^1^H-NMR (DMSO-*d*_6_): *δ* 8.57 (2H, d, J = 9.2 Hz, H-3′,5′), 8.52 (2H, d, J = 8.8 Hz, H-2′,6′), 7.73 (1H, d, J = 8.4 Hz, H-7), 7.63 (1H, s, H-4), 7.37 (1H, d, J = 8.4 Hz), 2.50 (3H, s, CH_3_). ^13^C-NMR (jmod, DMSO-*d*_6_): *δ* 149.13 (C), 146.57 (C), 135.75 (C), 133.82 (C), 131.98 (C), 130.62 (C), 128.91 (2CH), 127.32 (CH), 124.48 (2CH), 114.28 (CH), 113.76 (CH), 21.23 (CH_3_). ESI-MS (*m/z*): calcd. for C_14_H_11_N_3_O_2_ 254.092, found 254.092 [M+H]^+^.

#### 5.1.7. 5,6-Dimethyl-2-(4-nitrophenyl)-1H-benzo[d]imidazole (**7a**)

Compound **7a** (C_15_H_13_N_3_O_2_, MW 267.283) was obtained with a total yield of 62%; m.p. 260–262 °C; TLC (PE/EA 7/3): R_f_ 0.43. ^1^H-NMR (DMSO-*d*_6_): *δ* 8.38 (4H, s, H-2′,3′,5′,6′), 7.42 (2H, s, H-4,7), 2.34 (6H, s, 2CH_3_). ^13^C-NMR (DMSO-*d*_6_): *δ* 147.92 (C), 147.52 (C), 137.75 (C), 135.79 (2C), 132.07 (2C), 128.85 (2CH), 127.69 (2CH), 124.19 (2CH), 19.96 (2CH_3_). ESI-MS (*m/z*): calcd. for C_15_H_13_N_3_O_2_ 268.108, found 268.108 [M+H]^+^.

#### 5.1.8. 4-(1H-Benzo[d]imidazol-2-yl)aniline (**1b**)

Compound **1b** (C_13_H_11_N_3_, MW 209.247) was obtained with a total yield of 63%; m.p. >300 °C; TLC (CHCl_3_/CH_3_OH 95/5): R_f_ 0.38. ^1^H-NMR (DMSO-*d_6_*): δ 7.86 (2H, d, J = 8.4 Hz, H-2′,6′), 7.50 (2H, m, H-4,7), 7.14 (2H, m, H-5,6), 6.68 (2H, d, J = 8.4 Hz, H-3′,5′). ^13^C-NMR (jmod, DMSO-*d_6_*): δ 152.38 (C), 150.76 (2C), 138.88 (C), 127.85 (2CH), 121.46 (2CH), 116.63 (C), 113.54 (2CH), 113.22 (2CH). ESI-MS (*m/z*): calcd. for C_13_H_11_N_3_ 210.103, found 210.103 [M+H]^+^.

#### 5.1.9. 4-(5-Chloro-1H-benzo[d]imidazol-2-yl)aniline (**2b**)

Compound **2b** (C_13_H_10_ClN_3_, MW 243.692) was obtained with a total yield of 86%; m.p. 133–135 °C; TLC (CHCl_3_/CH_3_OH 95/5): R_f_ 0.29. ^1^H-NMR (DMSO-*d*_6_): *δ* 7.83 (2H, d, J = 8.4 Hz, H-2′,6′), 7.54 (1H, m, H-4), 7.43 (1H, ws, H-7), 7.13 (1H, d, J = 8.4 Hz, H-6), 6.66 (2H, d, J = 8.4 Hz), 5.66 (2H, s, NH_2_). ^13^C-NMR (APT, DMSO-*d*_6_+TFA-*d*): *δ* 153.63 (C), 150.93 (C), 132.17 (C), 130.24 (C), 129.65 (C), 129.60 (2CH), 125.52 (CH), 114.42 (CH), 114.13 (2CH), 112.76 (CH), 107.87 (C). ESI-MS (*m/z*): calcd. for C_13_H_10_ClN_3_ 244.064, found 244.064 [M+H]^+^.

#### 5.1.10. 4-(5,6-Dichloro-1H-benzo[d]imidazol-2-yl)aniline (**3b**)

Compound **3b** (C_13_H_9_Cl_2_N_3_, MW 278.137) was obtained with a total yield of 97%; m.p. 222–223 °C; TLC (CHCl_3_/CH_3_OH 95/5): R_f_ 0.28. ^1^H-NMR (DMSO-*d_6_*): δ 7.83 (2H, d, J = 7.6 Hz, H-2′6′), 7.78 (1H, s, H-4), 7.61 (1H, s, H-7), 6.67 (2H, d, J = 7.6 Hz, H-3′5′), 5.72 (2H, s, NH_2_). ^13^C-NMR (jmod, DMSO-*d_6_*): δ 155.21 (C), 151.22 (C), 143.89 (C), 134.56 (C), 128.12 (2CH), 123.46 (C), 123.28 (C), 118.78 (CH), 113.48 (2CH), 116.05 (C), 111.76 (CH). ESI-MS (*m/z*): calcd. for C_13_H_9_Cl_2_N_3_ 278.025, found 278.025 [M+H]^+^.

#### 5.1.11. 4-(5-Fluoro-1H-benzo[d]imidazol-2-yl)aniline (**4b**)

Compound **4b** (C_13_H_10_FN_3_, MW 227.237) was obtained with a total yield of 63%; m.p. 271 °C; TLC (PE/EA 7/3): R_f_ 0.12. ^1^H-NMR (DMSO-*d_6_*): δ 8.05 (2H, d, J = 8.8 Hz, H-2′,6′), 7.72 (1H, m, H-7), 7.54 (1H, m, H-4), 7.40 (1H, m, H-6), 6.78 (2H, d, J = 8.8 Hz, H-3′,5′). ^13^C-NMR (jmod, DMSO-*d_6_*): δ 160.81 (C), 158.42 (C), 153.90 (C), 151.15 (C), 132.42 (C), 129.67 (2CH), 128.48 (C), 114.43 (CH), 113.66 (2CH), 113.02 (CH), 99.95 (CH). ESI-MS (*m/z*): calcd. for C_13_H_10_FN_3_ 228.093, found 228.093 [M+H]^+^.

#### 5.1.12. 4-(5,6-Difluoro-1H-benzo[d]imidazol-2-yl)aniline (**5b**)

Compound **5b** (C_13_H_7_F_2_N_3_O_2_, MW 245.227) was obtained with a total yield of 87%; m.p. 246–247 °C; TLC (CHCl_3_/CH_3_OH 95/5): R_f_ 0.40. ^1^H-NMR (DMSO-*d*_6_): *δ* 7.81 (2H, d, J = 8.4 Hz, H-2′,6′), 7.54 (1H, s, H-4), 7.448 (1H, s, H-7), 6.67 (2H, d, J = 8.8 Hz, H-3′,5′), 5.64 (2H, s, NH2). ^13^C-NMR (jmod, DMSO-*d*_6_): *δ* 154.62 (C), 150.85 (2C), 146.21 (2C, dd, ^1^J_C-F_ = 236 Hz, ^2^J_C-F_ = 16 Hz, C-F), 127.75 (2CH), 116.53 (2C), 113.51 (2CH), 102.45-101.30 (2C, m, CH-F). ESI-MS (*m/z*): calcd. for C_13_H_9_F_2_N_3_ 246.083, found 246.083 [M+H]^+^.

#### 5.1.13. 4-(5-Methyl-1H-benzo[d]imidazol-2-yl)aniline (**6b**)

Compound **6b** (C_14_H_13_N_3_, MW 223.273) was obtained with a total yield of 31%; m.p. 110–112 °C; TLC (CHCl_3_/CH_3_OH 95/5): R_f_ 0.26. ^1^H-NMR (acetone-*d_6_*): *δ* 7.94 (2H, d, J = 8.8 Hz, H-2′,6′), 7.39 (1H, d, J = 8 Hz, H-7), 7.31 (1H, s, H-4), 6.97 (1H, d, J = 8 Hz, H-6), 6.78 (2H, d, J = 8.4 Hz, H-3′,5′), 5.09 (2H, s, NH_2_), 2.42 (3H, s, CH_3_). ^13^C-NMR (APT, DMSO-*d*_6_+ TFA-*d*): *δ* 153.69 (C), 149.49 (C), 135.10 (C), 131.54 (C), 129.33 (2CH), 129.11 (C), 126.45 (CH), 114.07 (2CH), 113.42 (2CH), 108.04 (C), 20.97 (CH_3_). ESI-MS (*m/z*): calcd. for C_14_H_13_N_3_ 224.118, found 224.118 [M+H]^+^.

#### 5.1.14. 4-(5,6-Dimethyl-1H-benzo[d]imidazol-2-yl)aniline (**7b**)

Compound **7b** (C_15_H_15_N_3_, MW 237.299) was obtained with a total yield of 88%; m.p. 109–111 °C; TLC (CHCl_3_/CH_3_OH 95/5): R_f_ 0.17. ^1^H-NMR (DMSO-*d*_6_): *δ* 7.80 (2H, d, J = 8.4 Hz, H-2′,6′), 7.31 (1H, s, H-7), 7.18 (1H, s, H-4), 6.65 (2H, d, J = 8.8 Hz, H-3′,5′), 5.55 (2H, s, NH_2_), 2.29 (6H, d, J = 7.2 Hz). ^13^C-NMR (jmod, DMSO-*d*_6_): *δ* 151.62 (C), 150.21 (C), 137.32 (2C), 130.01 (2C), 127.52 (2CH), 117.13 (2C), 114.37 (C), 113.78 (2CH), 19.84 (2CH_3_). ESI-MS (*m/z*): calcd. for C_15_H_15_N_3_ 238.134, found 238.134 [M+H]^+^.

### 5.2. Synthesis, Purification and Characterization of Benzimidazole Derivatives

#### 5.2.1. N-(4-(1H-Benzo[d]imidazol-2-yl)phenyl)-3,4,5-trimethoxybenzamide (**1c**), N-(4-(1H-Benzo[d]imidazol-2-yl)phenyl)-4-nitrobenzamide (**1d**) and N-(4-(1H-Benzo[d]imidazol-2-yl)phenyl)-4-chlorobenzamide (**1e**)

A mixture of 230 mg of 4-(1*H*-benzo[*d*]imidazol-2-yl)aniline (1.09 mmol) (**1b**) and 276 mg of 3,4,5-trimethoxybenzoyl chloride (1.20 mmol) (**9c**), or 222 mg of 4-nitrobenzoyl chloride (1.20 mmol) (**9d**) or 210 mg of 4-nitrobenzoyl chloride (1.20 mmol) (**9e**) (ratio 1:1.1) in 10 mL of DMF was stirred at 80 °C to obtain **1c**, **1d** and **1e**, respectively. The reactions were completed after 24 h. The reaction mixtures were poured into cold water, obtaining precipitation of the solid compounds. The precipitates were filtered off, washed with water and dried in an oven overnight. The pure products (**1c**, **1d** and **1e**) were obtained via crystallization using ethanol.

Compound **1c** (C_23_H_21_N_3_O_4_, MW 403.43) was obtained with a total yield of 20%; m.p. 164 °C; TLC (CHCl_3_/CH_3_OH 9.5/0.5): R_f_ 0.58. ^1^H-NMR (DMSO-*d*_6_): δ 10.61 (1H, s, NH), 8.30 (2H, d, J = 8.8 Hz, H-2′,6′), 8.09 (2H, d, J = 8.8 Hz, H-3′,5′), 7.75 (2H, m, H-4,7), 7.43 (2H, m, H-5,6), 8.09 (2H, d, J = 8.8, H-3′,5′), 7.35 (2H, s, H-2″,6″), 3.90 (6H, s, 2CH_3_-C3″,5″), 3.75 (3H, s, CH_3_-C4″). ^13^C-NMR (jmod, DMSO-*d*_6_): δ 165.34 (C), 152.64 (2C), 149.52 (C), 142.59 (C), 140.59 (C), 134.57 (C), 129.52 (2C), 128.15 (2CH), 124.35 (2CH), 120.52 (2CH), 120.39 (C), 114.18 (2CH), 105.54 (2CH), 60.12 (CH_3_), 56.16 (2CH_3_). ESI-MS (*m/z*): calcd. for C_23_H_21_N_3_O_4_ 404.160, found 404.161 [M+H]^+^.

Compound **1d** (C_20_H_14_N_4_O_3_, MW 358.35) was obtained with a total yield of 56%; m.p. >300 °C; TLC (CHCl_3_/CH_3_OH 9.5/0.5): R_f_ 0.49. ^1^H-NMR (DMSO-*d*_6_): *δ* 10.79 (1H, s, NH), 8.41 (2H, d, J = 8.8 Hz, H-3″,5″), 8.21 (2H, d, J = 2.8 Hz, H-2″,6″), 8.20 (2H, d, J = 8.8 Hz, H-3′,5′), 7.98 (2H, d, J = 8.8 Hz, H-2′,6′), 7.65 (1H, s, H-7), 7.53 (1H, s, H-4), 7.20 (2H, d, J = 5.2 Hz, H-5.6). ^13^C-NMR (jmod, DMSO-*d*_6_): *δ* 164.09 (C=O), 150.98 (C), 149.22 (C), 143.82 (C), 140.39 (C), 140.11 (C), 135.07 (C), 129.26 (2CH), 127.00 (2CH), 125.80 (C), 123.58 (2CH), 122.35 (CH), 121.59 (CH), 120.42 (2CH), 118.65 (CH), 111.15 (CH). ESI-MS (*m/z*): calcd. for C_20_H_14_N_4_O_3_ 359.114, found 359.114 [M+H]^+^.

Compound **1e** (C_20_H_14_ClN_3_O, MW 347.79) was obtained with a total yield of 62%; m.p. 170.5 °C; TLC (PE/EA 6/4): R_f_ 0.60. ^1^H-NMR (DMSO-*d*_6_): δ 8.36 (1H, s, NH), 8.20 (2H, d, J = 8.8 Hz, H-2′,6′), 8.02 (2H, m, H-3′,5′), 7.94 (2H, d, J = 8.4 Hz, H-2″,6″), 7.63 (2H, m, H-4,7), 7.56 (2H, d, J = 8.4 Hz, H-3″,5″), 7.24 (1H, m, H-5,6). ^13^C-NMR (jmod, DMSO-*d*_6_): δ 166.42 (C), 164.63 (C), 150.72 (C), 140.90 (C), 137.76 (C), 136.61 (C), 133.35 (C), 131.10 (2CH), 129.69 (2CH), 129.61 (C), 129.35 (CH), 129.24 (CH), 128.69 (CH), 128.49 (2CH), 127.23 (CH), 122.49 (CH), 120.33 (CH). ESI-MS (*m/z*): calcd. for C_20_H_14_ClN_3_O 348.090, found 348.090 [M+H]^+^.

#### 5.2.2. N-(4-(5-Chloro-1H-benzo[d]imidazol-2-yl)phenyl)-3,4,5-trimethoxybenzamide (**2c**), N-(4-(5-Chloro-1H-benzo[d]imidazol-2-yl)phenyl)-4-nitrobenzamide (**2d**) and N-(4-(5-Chloro-1H-benzo[d]imidazol-2-yl)phenyl)-4-chlorobenzamide (**2e**)

A mixture of 400 mg of 4-(5-chloro-1*H*-benzo[*d*]imidazol-2-yl)aniline (1.64 mmol) (**2b**) and 416 mg of 3,4,5-trimethoxybenzoyl chloride (1.80 mmol) (**9c**), or 334 mg of 4-nitrobenzoyl chloride (1.80 mmol) (**9d**) or 315 mg of 4-nitrobenzoyl chloride (1.80 mmol) (**9e**) (ratio 1:1.1) in 10 mL of DMF was stirred at 80 °C to obtain **2c**, **2d** and **2e**, respectively. The reactions were completed after 24 h. The reaction mixtures were poured into cold water, obtaining precipitation of the solid compounds. The precipitates were filtered off, washed with water and dried in an oven overnight. The crudes obtained were all purified using flash chromatography (CHCl_3_/CH_3_OH in a ratio of 96/4).

Compound **2c** (C_23_H_20_ClN_3_O_4_, MW 437.88) was obtained with a total yield of 15%; m.p. 286–287 °C; TLC (CHCl_3_/CH_3_OH 9.5/0.5): R_f_ 0.53. ^1^H-NMR (DMSO-*d*_6_): *δ* 10.35 (1H, s, NH), 8.17 (2H, d, J = 8 Hz, H-2′,6′), 7.95 (2H, d, J = 8.4 Hz, H-3′,5′), 7.70-7.64 (1H, m, H-6), 7.53 (1H, d, J = 8.4 Hz, H-7), 7.31 (2H, s, H-2″,6″), 7.24-7.19 (1H, m, H-4), 3.89 (6H, s, 2OCH_3_), 3.75 (3H, s, OCH_3_). ^13^C-NMR (APT, DMSO-*d*_6_+TFA-*d*): *δ* 165.66 (C=O), 152.63 (2C), 149.98 (C), 143.85 (C), 140.68 (C), 132.57 (C), 130.66 (C), 130.25 (C), 129.24 (C), 128.81 (2CH), 126.17 (CH), 120.64 (2CH), 117.19 (C), 115.22 (CH), 113.48 (CH), 105.38 (2CH), 60.03 (OCH_3_), 55.96 (2OCH_3_). ESI-MS (*m/z*): calcd. for C_23_H_20_ClN_3_O_4_ 438.121, found 438.122 [M+H]^+^.

Compound **2d** (C_20_H_13_ClN_4_O_3_, MW 392.80) was obtained with a total yield of 25%; m.p. >300 °C; TLC (CHCl_3_/CH_3_OH 9.5/0.5): R_f_ 0.45. ^1^H-NMR (DMSO-*d*_6_): *δ* 10.81 (1H, s, NH), 8.40 (2H, d, J = 8.8 Hz, H-3″,5″), 8.22 (2H, d, J = 8.8 Hz, H-2″,6″), 8.18 (2H, d, J = 8 Hz, H-3′,5′), 7.99 (2H, d, J = 8.8 Hz, H-2′,6′), 7.71-7.65 (1H, m, H-7), 7.54 (1H, d, J = 8.8 Hz, H-4), 7.25-7.20 (1H, m, H-6). ^13^C-NMR (APT, DMSO-*d*_6_+ TFA-*d*): *δ* 164.67 (C=O), 149.88 (C), 149.31 (C), 143.37 (C), 139.81 (C), 132.50 (C), 130.70 (C), 130.33 (C), 129.24 (2CH), 128.88 (2CH), 126.19 (CH), 123.49 (2CH), 120.64 (2CH), 117.61 (C), 115.25 (CH), 113.50 (CH). ESI-MS (*m/z*): calcd. for C_20_H_13_ClN_4_O_3_ 393.075, found 393.075 [M+H]^+^.

Compound **2e** (C_20_H_13_Cl_2_N_3_O, MW 382.24) was obtained with a total yield of 34%; m.p. >300 °C; TLC (CHCl_3_/CH_3_OH 9.5/0.5): R_f_ 0.62. ^1^H-NMR (APT, DMSO-*d*_6_+ TFA-*d*): *δ* 10.55 (1H, s, NH), 8.16 (2H, d, J = 8.8 Hz, H-3′,5′), 8.02 (2H, d, J = 8.8 Hz, H-2″,6″), 7.97 (2H, d, J = 8.8 Hz, H-2′,6′), 7.65 (2H, d, J = 8.4 Hz, H-3″,5″), 7.59 (2H, ws, H-4,7), 7.22 (1H, dd, J*_ortho_*= 8.6 Hz, J*_meta_*= 2 Hz, H-6). ^13^C-NMR (APT, DMSO-*d*_6_+ TFA-*d*): *δ* 165.16 (C=O), 149.93 (C), 143.81 (C), 136.97 (C), 132.88 (C), 132.50 (C), 130.67 (C), 130.30 (C), 129.67 (2CH), 128.88 (2CH), 128.49 (CH), 126.16 (CH), 120.47 (2CH), 119.25 (CH), 116.38 (C), 115.24 (CH), 113.54 (CH). ESI-MS (*m/z*): calcd. for C_20_H_13_Cl_2_N_3_O 382.051, found 382.051 [M+H]^+^.

#### 5.2.3. N-(4-(5,6-Dichloro-1H-benzo[d]imidazol-2-yl)phenyl)-3,4,5-trimethoxybenzamide (**3c**), N-(4-(5,6-Dichloro-1H-benzo[d]imidazol-2-yl)phenyl)-4-nitrobenzamide (**3d**), N-(4-(5,6-Dichloro-1H-benzo[d]imidazol-2-yl)phenyl)-4-chlorobenzamide (**3e**)

A mixture of 300 mg of 4-(5,6-dichloro-1*H*-benzo[*d*]imidazol-2-yl)aniline (1.08 mmol) (**3b**) and 274 mg of 3,4,5-trimethoxybenzoyl chloride (1.19 mmol) (**9c**), or 220 mg of 4-nitrobenzoyl chloride (1.19 mmol) (**9d**), or 208 mg of 4-nitrobenzoyl chloride (1.19 mmol) (**9e**) [ratio 1:1.1] in 10 ml of DMF was stirred at 80 °C, to obtain **3c**, **3d**, **3e** respectively. The reactions were completed after 24 h. The reaction mixtures were poured into cold water, obtaining precipitation of the solid compounds. The precipitates were filtered off, washed with water and dried in oven overnight. The pure products were obtained by crystallisation from ethanol (**3c** and **3e**) or from methanol (**3d**). 

Compound **3c** (C_23_H_19_Cl_2_N_3_O_4_, MW 472.32) was obtained in total yield 32%; m.p. 150.6 °C; TLC (PS/EA 6/4): R_f_ 0.34. ^1^H-NMR (DMSO-*d*_6_): δ 10.43 (1H, s, NH), 8.13 (2H, d, J = 8.8 Hz), 7.95 (2H, d, J = 8.4 Hz), 7.83 (2H, s, H-4,7), 7.29 (2H, s, H-2′,6′), 3.87 (3H, s, CH_3_), 3.81 (6H, ws, 2CH_3_). ^13^C-NMR (jmod, DMSO-*d*_6_): δ 166.94 (C), 165.29 (C), 153.61 (C), 152.62 (C), 152.59 (C), 141.32 (C), 141.20 (C), 140.45 (C), 129.63 (C), 127.32 (2CH), 125.81 (C), 124.50 (C), 124.00 (C), 120.58 (2CH), 113.75 (2CH), 105.30 (2CH), 60.10 (CH_3_), 55.88 (2CH_3_). ESI-MS (*m/z*): calcd for C_23_H_19_Cl_2_N_3_O_4_ 472.083, found 472.082 [M+H]^+^. 

Compound **3d** (C_20_H_12_Cl_2_N_4_O_3_, MW 427.24) was obtained in total yield 10%; m.p. >300 °C; TLC (CHCl_3_/CH_3_OH 9.5/0.5): R_f_ 0.53. ^1^H-NMR (DMSO-*d*_6_): *δ* 10.83 (1H, s, NH), 8.41 (2H, d, J = 8.8 Hz, H-3″,5″), 8.22 (2H, d, J = 8.8 Hz, H-2″,6″), 8.19 (2H, d, J = 8.8 Hz, H-3′,5′), 8.00 (2H, d, J = 8.8 Hz, H-2′,6′), 7.93 (1H, s, H-4), 7.76 (1H, s, H-7). ^13^C-NMR (APT, DMSO-*d*_6_, TFA-*d*): *δ* 164.57 (C=O), 153.64 (C), 151.60 (C), 149.32 (C), 142.96 (C), 139.94 (C), 133.20 (C), 131.48 (C), 129.92 (CH), 129.29 (2CH), 128.73 (CH), 127.62 (C), 123.53 (2CH), 120.56 (CH), 119.21 (C), 115.54 (CH), 114.56 (CH), 114.11 (CH). ESI-MS (*m/z*): calcd for C_20_H_12_Cl_2_N_4_O_3_ 427.036, found 427.036 [M+H]^+^. 

Compound **3e** (C_20_H_12_Cl_3_N_3_O, MW 416.69) was obtained in total yield 23%; m.p. 161–162 °C; TLC (PS/EA 6/4): R_f_ 0.69. ^1^H-NMR (DMSO-*d*_6_): δ 10.63 (1H, s, NH), 8.17 (2H, d, J = 8.4 Hz, H-2′,6′), 7.86 (2H, s, H-4,7), 7.81 (2H, d, J = 8.4 Hz, H-3′,5′), 7.62 (2H, d, J = 8.4 Hz, H-2″-6″), 7.55 (2H, d, J = 8.4 Hz, H-3″,5″). ^13^C-NMR (jmod, DMSO-*d*_6_): δ 166.47 (C), 164.82 (C), 153.22 (C), 141.47 (C), 137.78 (2C), 136.68 (C), 133.20 (C), 131.10 (2CH), 129.66 (CH), 129.53 (C), 128.69 (2CH), 128.50 (CH), 127.56 (CH), 124.97 (C), 120.34 (CH), 115.94 (CH), 113.90 (CH). ESI-MS (*m/z*): calcd for C_20_H_12_Cl_3_N_3_O 416.012, found 416.012 [M+H]^+^.

#### 5.2.4. N-(4-(5-Fluoro-1H-benzo[d]imidazol-2-yl)phenyl)-3,4,5-trimethoxybenzamide (**4c**), N-(4-(5-Fluoro-1H-benzo[d]imidazol-2-yl)phenyl)-4-nitrobenzamide (**4d**), N-(4-(5-Fluoro-1H-benzo[d]imidazol-2-yl)phenyl)-4-chlorobenzamide (**4e**)

A mixture of 500 mg of 4-(5-fluoro-1*H*-benzo[*d*]imidazol-2-yl)aniline (2.20 mmol) (**4b**) and 558 mg of 3,4,5-trimethoxybenzoyl chloride (2.42 mmol) (**9c**), or 449 mg of 4-nitrobenzoyl chloride (2.42 mmol) (**9d**), or 424 mg of 4-nitrobenzoyl chloride (2.42 mmol) (**9e**) [ratio 1:1.1] in 10 mL of DMF was stirred at 80 °C, to obtain **4c**, **4d**, **4e** respectively. The reactions were completed after 24 h. The reaction mixtures were poured into cold water, obtaining precipitation of the solid compounds. The precipitates were filtered off, washed with water and dried in oven overnight. The crudes obtained were purified by flash chromatography (CHCl_3_/CH_3_OH in ratio 95/5) for **4d**. Pure **4c** and **4e** were obtained by crystallisation from ethanol. 

Compound **4c** (C_23_H_20_FN_3_O_4_, MW 421.42) was obtained in total yield 13%; m.p. 171.8 °C; TLC (CHCl_3_/CH_3_OH 9.5/0.5): R_f_ 0.46. ^1^H-NMR (DMSO-*d*_6_): δ 10.60 (1H, s, NH), 8.37 (2H, s, H-2″,6″), 8.16 (2H, d, J = 8.4 Hz, H-2′6′), 7.80 (2H, d, J = 8.4, H-3′,5′), 7.63 (1H, m, H-6), 7.43 (1H, m, H-4), 7.14 (1H, m, H-7). ^13^C-NMR (jmod, DMSO-*d*_6_): δ 165.19 (C), 158.94 (1C, ^1^J_C-F_ = 235 Hz, C-F), 151.67 (2C), 140.91 (C), 140.52 (2C), 137.86 (C), 133.78 (C), 128.21 (CH), 127.75 (2CH), 122.96 (2C), 122.80 (C), 120.48 (2CH), 117.20 (CH), 115.29 (1C, ^3^J_C-F_ = 10 Hz, CH), 111.05 (1C, ^2^J_C-F_ = 25 Hz, CH), 100.85 (1C, ^2^J_C-F_ = 26 Hz, CH), 60.10 (CH_3_)¸ 56.13 (2CH_3_). ESI-MS (*m/z*): calcd for C_23_H_20_FN_3_O_4_ 422.151, found 422.151 [M+H]^+^. 

Compound **4d** (C_20_H_13_FN_4_O_3_, MW 376.34) was obtained in total yield 29%; m.p. >300 °C; TLC (CHCl_3_/CH_3_OH 9.5/0.5): R_f_ 0.35. ^1^H-NMR (DMSO-*d*_6_): *δ* 10.79 (1H, s, NH), 8.4 (2H, d, J = 8.8 Hz, H-3″,5″), 8.22 (2H, d, J = 8.8 Hz, H-2″,6″), 8.17 (2H, d, J = 8.4 Hz, H-3′,5′), 7.98 (2H, d, J = 8.8 Hz, H-2′,6′), 7.58 (1H, ws, H-7), 7.38 (1H, ws, H-4), 7.06 (1H, m, H-6). ^13^C-NMR (APT, DMSO-*d*_6_+TFA-*d*): *δ* 164.59 (C=O), 160.11 (C, d, ^1^J = 240 Hz, C-F), 149.93 (C), 149.32 (C), 143.29 (C), 139.84 (C), 132.23 (C, m, C-F), 129.28 (2CH), 128.79 (2CH), 128.36 (C), 123.52 (2CH), 120.60 (2CH), 115.34 (C, d, ^3^J_C-F_ = 10 Hz, CH-7), 114.41 (C, d, ^2^J_C-F_= 25 Hz, CH-6), 100.47 (C, d, ^2^J_C-F_ = 29 Hz, CH-4). ESI-MS (*m/z*): calcd for C_20_H_13_FN_4_O_3_ 377.104, found 377.104 [M+H]^+^. 

Compound **4e** (C_20_H_13_ClFN_3_O, MW 365.79) was obtained in total yield 37%; m.p. 138.6 °C; TLC (CHCl_3_/CH_3_OH 9.5/0.5): R_f_ 0.66. ^1^H-NMR (DMSO-*d*_6_): δ 10.55 (1H, s, NH), 8.15 (1H, m, H-6), 7.98-7.93 (4H, m, H-3’,5’,3″,5″), 7.65-7.56 (4H, m, H-2′,6′,2″,6″), 7.39 (1H, m, H-4), 7.07 (1H, m, H-7). ^13^C-NMR (jmod, DMSO-*d*_6_): δ 165.55 (C), 159.49 (1C, d, ^1^J_C-F_ = 234 Hz, C-F), 153.39 (C), 141.51 (2C), 137.49 (2C), 134.31 (C), 132.02 (CH), 130.58 (2CH), 129.62 (2CH), 129.41 (2CH), 126.00 (C), 121.23 (2CH), 110.88 (1C, d, ^2^J_C-F_ = 25 Hz, CH). ESI-MS (*m/z*): calcd for C_20_H_13_ClFN_3_O 366.080, found 366.081 [M+H]^+^.

#### 5.2.5. N-(4-(5,6-Difluoro-1H-benzo[d]imidazol-2-yl)phenyl)-3,4,5-trimethoxybenzamide (**5c**), N-(4-(5,6-Difluoro-1H-benzo[d]imidazol-2-yl)phenyl)-4-nitrobenzamide (**5d**), N-(4-(5,6-Difluoro-1H-benzo[d]imidazol-2-yl)phenyl)-4-chlorobenzamide (**5e**)

A mixture of 300 mg of 4-(5,6-difluoro-1*H*-benzo[*d*]imidazol-2-yl)aniline (1.22 mmol) (**5b**) and 309 mg of 3,4,5-trimethoxybenzoyl chloride (1.34 mmol) (**9c**), or 249 mg of 4-nitrobenzoyl chloride (1.34 mmol) (**9d**), or 235 mg of 4-nitrobenzoyl chloride (1.34 mmol) (**9e**) [ratio 1:1.1] in 10 mL of DMF was stirred at 80 °C, to obtain **5c**, **5d**, **5e** respectively. The reactions were completed after 24 h. The reaction mixtures were poured into cold water, obtaining precipitation of the solid compounds. The precipitates were filtered off, washed with water and dried in oven overnight. The crudes obtained were purified by flash chromatography (CHCl_3_/CH_3_OH in ratio 98/2) for **5d** and **5e**. Pure **5c** was obtained by crystallisation from methanol. 

Compound **5c** (C_23_H_19_F_2_N_3_O_4_, MW 439.41) was obtained in total yield 28%; m.p. 285–287 °C; TLC (CHCl_3_/CH_3_OH 9.5/0.5): R_f_ 0.50. ^1^H-NMR (DMSO-*d*_6_): *δ* 10.34 (1H, s, NH), 8.14 (2H, d, J = 8.4 Hz, H-2′,6′), 7.94 (2H, d, J = 8.4 Hz, H-3′,5′), 7.69 (1H, m, H-7), 7.55 (1H, m, H-4), 7.30 (2H, s, H-2″,6″), 3.89 (6H, s, C-3″,5″-OCH_3_), 3.75 (3H, s, C-4″-OCH_3_). ^13^C-NMR (jmod, DMSO-*d*_6_): *δ* 165.10 (C=O), 153.17 (C), 152.63 (2C), 146.64 (2C, dd, ^1^J_C-F_ = 251 Hz, ^2^J_C-F_ = 14 Hz, C-F), 140.78 (C), 140.45 (C), 139.28 (C, d, J_C-F_ = 12 Hz, C), 130.34 (C, d, J_C-F_ = 12 Hz, C), 129.78 (C), 126.94 (2CH), 124.76 (C), 120.47 (2CH), 105.87 (C, d, J_C-F_ = 19 Hz, CH-F), 105.37 (2CH), 99.17 (C, d, J_C-F_ = 22 Hz, CH-F), 60.12 (CH_3_), 56.11 (2CH_3_). ESI-MS (*m/z*): calcd for C_23_H_19_F_2_N_3_O_4_ 440.142, found 440.142 [M+H]^+^. 

Compound **5d** (C_20_H_12_F_2_N_4_O_3_, MW 394.33) was obtained in total yield 48%; m.p. >300 °C; TLC (CHCl_3_/CH_3_OH 9.5/0.5): R_f_ 0.37. ^1^H-NMR (DMSO-*d*_6_): *δ* 10.79 (1H, s, NH), 8.40 (2H, d, J = 8.8 Hz, H-3″,5″), 8.22 (2H, d, J = 8.8 Hz, H-2″,6″), 8.16 (2H, d, J = 8.8 Hz, H-3′,5′), 7.98 (2H, d, J = 8.8 Hz, H-2′,6′), 7.64 (2H, ws, H-4,7). ^13^C-NMR (APT, DMSO-*d*_6_+ TFA-*d*): *δ* 164.55 (C=O), 150.62 (C), 149.34 (C), 148.71 (2C, dd, ^1^J_C-F_ = 245 Hz, ^2^J_C-F_ = 17 Hz, C-F), 143.34 (C), 139.88 (C), 138.42 (C), 129.25 (2CH), 128.78 (2CH), 127.76 (C, m, C), 123.44 (2CH), 120.59 (2CH), 117.87 (C, m, C), 102.50 (2C, dd, ^2^J = 15.5 Hz, ^3^J = 7 Hz, CH). ESI-MS (*m/z*): calcd for C_20_H_12_F_2_N_4_O_3_ 395.095, found 395.095 [M+H]^+^. 

Compound **5e** (C_20_H_12_ClF_2_N_3_O, MW 383.78 °C) was obtained in total yield 47%; m.p. >300 °C; TLC (CHCl_3_/CH_3_OH 9.5/0.5): R_f_ 0.41. ^1^H-NMR (DMSO-*d*_6_): *δ* 10.54(1H, s, NH), 8.14 (1H, d, J = 8.8 Hz, H-7), 8.02 (2H, d, J = 8.4 Hz, H-2′,6′), 7.954 (2H, m, H-4, 2″,6″), 7.64 (2H, d, J = 8.4 Hz, H-3′,5′), 7.57 (2H, d, J = 8.4 Hz, H-3″,5″). ^13^C-NMR (jmod, DMSO-*d*_6_): *δ* 166.49 (C=O), 164.70 (C) 153.18 (C), 146.76 (2C, dd, ^1^J_C-F_ = 237 Hz, ^2^J_C-F_ = 14 Hz, C-F), 140.72 (C), 137.77 (C), 136.63 (C), 133.43 (C), 131.15 (2CH), 129.73 (2CH), 128.74 (2CH), 128.55 (2CH), 127.01 (CH), 124.93 (C), 120.37 (CH). ESI-MS (*m/z*): calcd for C_20_H_12_ClF_2_N_3_O 384.071, found 384.071 [M+H]^+^.

#### 5.2.6. N-(4-(5-Methyl-1H-benzo[d]imidazol-2-yl)phenyl)- 3,4,5-trimethoxybenzamide (**6c**), N-(4-(5-Methyl-1H-benzo[d]imidazol-2-yl)phenyl)-4-nitrobenzamide (**6d**), N-(4-(5-Methyl-1H-benzo[d]imidazol-2-yl)phenyl)-4-chlorobenzamide (**6e**)

A mixture of 350 mg of 4-(5-methyl-1*H*-benzo[*d*]imidazol-2-yl)aniline (1.57 mmol) (**6b**) and 399 mg of 3,4,5-trimethoxybenzoyl chloride (1.73 mmol) (**9c**), or 321 mg of 4-nitrobenzoyl chloride (1.73 mmol) (**9d**), or 303 mg of 4-nitrobenzoyl chloride (1.73 mmol) (**9e**) [ratio 1:1.1] in 10 mL of DMF was stirred at 80 °C, to obtain **6c**, **6d**, **6e** respectively. The reactions were completed after 24 h. The reaction mixtures were poured into cold water, obtaining precipitation of the solid compounds. The precipitates were filtered off, washed with water and dried in oven overnight. The solid obtained were pure for derivatives **6d** and **6e**. Crude of **6c** was purified by flash chromatography (CHCl_3_/CH_3_OH in ratio 95/5). 

Compound **6c** (C_24_H_23_N_3_O_4_, MW 417.46) was obtained in total yield 35%; m.p. 287-279 °C; TLC (CHCl_3_/CH_3_OH 9.5/0.5): R_f_ 0.50. ^1^H-NMR (DMSO-*d*_6_): *δ* 10.32 (1H, s, NH), 8.15 (2H, d, J = 8.8 Hz, H-3′,5′), 7.93 (2H, d, J = 8.8 Hz, H-2′,6′), 7.52-7.50 (1H, m, H-7), 7.43-7.39 (1H, m, H-4), 7.31 (2H, s, H-2″,6″), 7.02-7.00 (1H, m, H-6), 3.90 (6H, s, 2OCH_3_), 3.75 (3H, s, OCH_3_), 2.44 (3H, s, CH_3_). ^13^C-NMR (APT, DMSO-*d*_6_+ TFA-*d*): *δ* 165.49 (C=O), 152.65 (2C), 148.24 (C), 143.68 (C), 140.73 (C), 136.11 (C), 131.69 (C), 129.50 (C), 129.28 (C), 128.62 (2CH), 127.40 (CH), 120.57 (2CH), 120.47 (CH), 117.28 (C), 113.06 (CH), 105.44 (2CH), 59.99 (OCH_3_), 55.97 (2OCH_3_), 20.94 (CH_3_). ESI-MS (*m/z*): calcd for C_24_H_23_N_3_O_4_ 418.176, found 418.176 [M+H]^+^. 

Compound **6d** (C_21_H_16_N_4_O_3_, MW 372.38) was obtained in total yield 80%; m.p. 297–298 °C; TLC (CHCl_3_/CH_3_OH 9.5/0.5): R_f_ 0.45. ^1^H-NMR (DMSO-*d*_6_): *δ* 10.77 (1H, s, NH), 8.40 (2H, d, J = 8.8 Hz, H-3″,5″), 8.22 (2H, d, J = 8.8 Hz, H-2″,6″), 8.17 (2H, d, J = 8.8 Hz, H-3′,5′), 7.97 (2H, d, J = 8.8 Hz, H-2′,6′), 7.47 (1H, d, J = 8 Hz, H-7), 7.37 (1H, s, H-4), 7.03 (1H, d, J = 8 Hz, H-6), 2.44 (3H, s, CH_3_). ^13^C-NMR (APT, DMSO-*d*_6_+ TFA-*d*): *δ* 164.48 (C=O), 149.35 (C), 148.11 (C), 143.23 (C), 139.87 (C), 136.15 (C), 131.69 (C), 129.50 (C), 129.29 (2CH), 128.72 (2CH), 127.44 (CH), 123.49 (2CH), 120.58 (2CH), 117.80 (C), 113.29 (CH), 113.10 (CH), 20.93 (CH_3_). ESI-MS (*m/z*): calcd for C_21_H_16_N_4_O_3_ 373.130, found 373.130 [M+H]^+^. 

Compound **6e** (C_21_H_16_ClN_3_O, MW 361.82) was obtained in total yield 60%; m.p. >300 °C; TLC (CHCl_3_/CH_3_OH 9.5/0.5): R_f_ 0.82. ^1^H-NMR (DMSO-*d*_6_): *δ* 10.52 (1H, s, NH), 8.14 (2H, d, J = 8.4 Hz, H-2″,6″), 8.02 (2H, d, J = 8 Hz, H-2′,6′), 7.95 (2H, d, J = 8.4 Hz, H-3′,5′), 7.64 (2H, d, J = 8.4 Hz, H-3″,5″), 7.46 (1H, d, J = 7.6 Hz, H-7), 7.36 (1H, s, H-4), 7.02 (1H, d, J = 8 Hz, H-6), 2.43 (3H, s, CH_3_). ^13^C-NMR (APT, DMSO-*d*_6_+TFA-*d*): *δ* 165.08 (C=O), 148.21 (C), 143.52 (C), 136.94 (C), 136.11 (C), 132.93 (C), 131.71 (C), 129.68 (2CH), 129.52 (C), 128.65 (2CH), 128.49 (2CH), 127.41 (CH), 120.46 (CH), 117.44 (C), 113.27 (CH), 113.08 (CH), 20.95 (CH_3_). ESI-MS (*m/z*): calcd for C_21_H_16_ClN_3_O 362.105, found 362.105 [M+H]^+^.

#### 5.2.7. N-(4-(5,6-Dimethyl-1H-benzo[d]imidazol-2-yl)phenyl)-3,4,5-trimethoxybenzamide (**7c**), N-(4-(5,6-Dimethyl-1H-benzo[d]imidazol-2-yl)phenyl)-4-nitrobenzamide (**7d**), N-(4-(5,6-Dimethyl-1H-benzo[d]imidazol-2-yl)phenyl)-4-chlorobenzamide (**7e**)

A mixture of 300 mg of 4-(5,6-dimethyl-1*H*-benzo[*d*]imidazol-2-yl)aniline (1.26 mmol) (**7b**) and 321 mg of 3,4,5-trimethoxybenzoyl chloride (1.39 mmol) (**9c**), or 258 mg of 4-nitrobenzoyl chloride (1.39 mmol) (**9d**), or 243 mg of 4-nitrobenzoyl chloride (1.39 mmol) (**9e**) [ratio 1:1.1] in 10 mL of DMF was stirred at 80 °C, to obtain **7c**, **7d**, **7e** respectively. The reactions were completed after 24 h. The reaction mixtures were poured into cold water, obtaining precipitation of the solid compounds. The precipitates were filtered off, washed with water and dried in oven overnight. The pure solid products **7c**, **7d**, **7e** were obtained by washing the crudes with few millilitres of diethyl ether. 

Compound **7c** (C_25_H_25_N_3_O_4_, MW 431.48) was obtained in total yield 40%; m.p. 272–274 °C; TLC (CHCl_3_/CH_3_OH 9.5/0.5): R_f_ 0.47. ^1^H-NMR (DMSO-*d*_6_): *δ* 10.43 (1H, s, NH), 8.16 (2H, d, J = 8.4 Hz, H-2′,6′), 8.00 (2H, d, J = 8.4 Hz, H-3′,5′), 7.46 (2H, s, H-4,7), 7.32 (2H, s, H-2″,6″), 3.89 (6H, s, 2OCH_3_), 3.75 (3H, s, OCH_3_), 2.37 (6H, s,2CH_3_). ^13^C-NMR (jmod, DMSO-*d*_6_): *δ* 165.20 (C=O), 152.65 (2C), 149.05 (C), 141.66 (C), 140.56 (C), 134.42 (C), 132.63 (2C), 129.62 (2C), 127.51 (2CH), 122.04 (C), 120.50 (2CH), 114.24 (2CH), 105.42 (2CH), 60.13 (OCH_3_), 56.13 (2OCH_3_), 19.94 (2CH_3_). ESI-MS (*m/z*): calcd for C_25_H_25_N_3_O_4_ 432.192, found 432.192 [M+H]^+^. 

Compound **7d** (C_22_H_18_N_4_O_3_, MW 386.40) was obtained in total yield 18%; m.p. >300 °C; TLC (CHCl_3_/CH_3_OH 9.5/0.5): R_f_ 0.40. ^1^H-NMR (DMSO-*d*_6_): *δ* 10.77 (1H, s, NH), 8.40 (2H, d, J = 8.8 Hz, H-3″,5″), 8.34 (2H, d, J = 2 Hz, H-2″,6″), 8.23-8.14 (4H, m, H-2′,3′,5′,6′), 7.96 (1H, d, J = 8.8 Hz, H-7), 7.35 (1H, s, H-4), 2.33 (6H, s, 2CH_3_). ^13^C-NMR (jmod, DMSO-*d*_6_): *δ* 165.77 (C), 164.03 (C), 150.05 (C), 149.21 (C), 140.41 (C), 139.84 (C), 136.36 (C), 132.77 (C), 130.67 (2CH), 130.48 (C), 129.25 (2CH), 126.77 (CH), 125.97 (C), 123.71 (2CH), 123.57 (2CH), 120.38 (C), 19.99 (2CH_3_). ESI-MS (*m/z*): calcd for C_22_H_18_N_4_O_3_ 387.145, found 387.145 [M+H]^+^. 

Compound **7e** (C_22_H_18_ClN_3_O, MW 375.85) was obtained in total yield 19%; m.p. >300 °C; TLC (CHCl_3_/CH_3_OH 9.5/0.5): R_f_ 0.32. ^1^H-NMR (DMSO-*d*_6_): *δ* 10.54 (1H, s, NH), 8.14-7.93 (6H, m, H-2′, 3′, 5′, 6′, 2″, 6″), 7.65-7.56 (3H, m, H-3″, 5″, 7), 7.37 (1H, s, H-4), 2.34 (6H, s, 2CH_3_). ^13^C-NMR (jmod, DMSO-*d*_6_): *δ* 166.43 (C), 164.63 (C), 149.89 (C), 140.43 (C), 137.76 (C), 136.58 (C), 133.39 (C), 131.11 (2CH), 130.92 (C), 129.66 (2CH), 129.59 (C), 128.72 (2CH), 128.51 (2CH), 126.88 (CH), 124.86 (C), 120.30 (CH), 19.97 (2CH_3_). ESI-MS (*m/z*): calcd for C_22_H_18_ClN_3_O 376.121, found 376.121 [M+H]^+^.

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
