# Peer review of "Benzimidazole-2-Phenyl-Carboxamides as Dual-Target Inhibitors of BVDV Entry and Replication"

_viruses, 2022, doi:10.3390/v14061300_

Round 1
Reviewer 1 Report
- This is a straight forward test of benzimidazole compounds for antiviral activity against cytopathic (cp) NADL BVDV in vitro. Overall the manuscript is well-written and organized. There are a few syntactical issues that can easily be resolved by the editor.
- The antiviral assay and cell culture methods are standard methods used to characterize antiviral activity; however, important details are missing such as the number of (technical) replicate wells for each compound and concentration and the number of times the assays were repeated (independent experiments).
- It is important to include the method used for data analysis including statistical tests and p values.
- In Figure 2, the data points lack SD or SEM bars (see comment above about statistics). The inclusion of SD or SEM on the graphs is especially important since the differences between the controls and test compounds appears to be small.
- If the intent is to use these compounds on BVDV-infected cattle, it would be better to have tested their antiviral properties on noncytopathic (ncp) BVDVs as opposed to cytopathic NADL. BVDVs present in nature are predominantly of the ncp biotype and are responsible for the majority of economic losses to cattle producers.
- If the goal is to treat cattle with these compounds, then a discussion of potential toxicity, cost of these compounds and cost of treatments would be important.
- The discussion should include a comparison with the findings of other investigators that have examined benzimidazole compounds antiviral activity against BVDV (or HCV). These journal articles should be cited.
- The discussion should include the potential confounding effects of benzimidazoles on cell biology which might interfere with interpretation of cytotoxicity assays such as effects on tubulin polymerization arresting cells in G2/M phase and reducing cell division, effects on glucose utilization inhibiting cell growth and apoptosis which might interfere with cell viability in the antiviral assay.
Author Response
Dear Reviewer,
We would like to thank you for the comments that stimulated us to greatly improve our draft. We addressed all your concerns and replied point by point at your comments, as follow.
- This is a straight forward test of benzimidazole compounds for antiviral activity against cytopathic (cp) NADL BVDV in vitro. Overall the manuscript is well-written and organized. There are a few syntactical issues that can easily be resolved by the editor.
- We tried to fix all the syntactical issues.
- The antiviral assay and cell culture methods are standard methods used to characterize antiviral activity; however, important details are missing such as the number of (technical) replicate wells for each compound and concentration and the number of times the assays were repeated (independent experiments).
- Each experiment was conducted in duplicate (two wells in parallel) and the experiments were performed in three copies.
- It is important to include the method used for data analysis including statistical tests and p values.
- The MTT data were processed using the statistical method of linear regression, while the TOA was read using the plaque reduction method. Draft was implemented accordingly.
- In Figure 2, the data points lack SD or SEM bars (see comment above about statistics). The inclusion of SD or SEM on the graphs is especially important since the differences between the controls and test compounds appears to be small.
- Figure 2 was replaced with bar graphs (as suggested by Reviewer 2) and SEM bars were also added.
- If the intent is to use these compounds on BVDV-infected cattle, it would be better to have tested their antiviral properties on noncytopathic (ncp) BVDVs as opposed to cytopathic NADL. BVDVs present in nature are predominantly of the ncp biotype and are responsible for the majority of economic losses to cattle producers.
- Our intent focused on the discovery and selection of potent and selective compounds capable of inhibiting virus infection and replication. Next step will be the optimization of leads compound in order to obtain a drug-like compound. At that stage we will test the compounds against ncp BVDV too.
- If the goal is to treat cattle with these compounds, then a discussion of potential toxicity, cost of these compounds and cost of treatments would be important.
- It is going to be the latest goal, so at this stage the costs for the synthesis are almost neglectable.
- The discussion should include a comparison with the findings of other investigators that have examined benzimidazole compounds antiviral activity against BVDV (or HCV). These journal articles should be cited.
- The journal articles were cited as suggested.
- The discussion should include the potential confounding effects of benzimidazoles on cell biology which might interfere with interpretation of cytotoxicity assays such as effects on tubulin polymerization arresting cells in G2/M phase and reducing cell division, effects on glucose utilization inhibiting cell growth and apoptosis which might interfere with cell viability in the antiviral assay.
- The potential and eventual cytotoxic mechanism of action is negligible when CC50 values are higher than 100 µM and/or the selectivity index ranges between 300 and 500.

Reviewer 2 Report
The manuscript by Roberta Ibba and colleagues tested and characterized a new series of compounds emerged from an isosteric substitution of main scaffold in previously reported anti-BVDV compounds. The authors reported three compounds inhibiting early stages of virus infection and showed binding of these three compounds and their derivatives to viral E2 protein with in silico docking. The authors also predicted one compound, ligand 2c, binds to RdRp weaker than to E2. The study is important for antiviral development targeting BVDV and of interest to the readers of Viruses. Some questions need to be addressed to make the study acceptable to publish.
Major points:
The antiviral mechanistic study was designed only for drug treatment of cells but missed critical control experiments with drug treatment with virus before infection. If the compounds inhibit the virus infection by binding to E2, they should be expected to inhibit virus infection by treating the virus alone.
The authors should explore how the compounds binding to the pocket on E2 affects the viral infectivity. Do they inhibit conformational changes of E2 required for viral fusion or inhibit receptor binding?
Minor points:
Page 4, Lines 147-150: units for volume should be changed from mm3 and dm3 to ml and l.
Figure 2, a better presentation for the drug addition assay can be bar graph for each compound plot against drug addition time spans.
The authors should discuss why EC50 of 2c, 6c and 7c are lower than NM108 and Ribavirin but reductions of virus titer in 2c, 6c and 7c treated samples are smaller than NM108 and Ribavirin.
Author Response
Dear Reviewer,
We would like to thank you for the comments that stimulated us to greatly improve our draft. We addressed all your concerns and replied point by point at your comments, as follow.
Major points:
The antiviral mechanistic study was designed only for drug treatment of cells but missed critical control experiments with drug treatment with virus before infection. If the compounds inhibit the virus infection by binding to E2, they should be expected to inhibit virus infection by treating the virus alone.
The virucidal assay was performed, as suggested.
The authors should explore how the compounds binding to the pocket on E2 affects the viral infectivity. Do they inhibit conformational changes of E2 required for viral fusion or inhibit receptor binding?
Since the single treatment of virus or cell pretreatment does not results in the highest antiviral activity, while it is proved when the three components, virus, cell and ligand are mixed at the same time, it would suggest that compounds inhibit the receptor binding.
Minor points:
Page 4, Lines 147-150: units for volume should be changed from mm3 and dm3 to ml and l.
They were changed accordingly.
Figure 2, a better presentation for the drug addition assay can be bar graph for each compound plot against drug addition time spans.
Figure 2 was replaced with bar graph representation with statistical analysis also reported, as kindly suggested.
The authors should discuss why EC50 of 2c, 6c and 7c are lower than NM108 and Ribavirin but reductions of virus titer in 2c, 6c and 7c treated samples are smaller than NM108 and Ribavirin
The slight differences in the two assays are due to the two different detection methods used, MTT and plaque reduction assays.
